# Effects of Ultrasonication and Modified Atmosphere Packaging on the Physicochemical Characteristics and Quality of Ready-to-Eat Pomegranate Arils

**Farid Moradinezhad** [1,*] **, Asma Heydari** [1] **and Elham Ansarifar** [2]

[1] Department of Horticultural Science, Faculty of Agriculture, University of Birjand,
    Birjand P.O. Box 97175-331, Iran; asm.haidari.1990@gmail.com
[2] Social Determinants of Health Research Center, Department of Public Health, School of Health, Birjand
    University of Medical Science, Birjand P.O. Box 97175-379, Iran; ansarifar.elham@gmail.com
[*] Correspondence: fmoradinezhad@birjand.ac.ir or fmn_46@yahoo.com

**Abstract:** The demand for ready-to-eat pomegranate arils has increased due to the high nutritional value and quality of this valuable fruit. However, the shelf life of arils is short. Therefore, we examined the effect of ultrasonication treatment (35 kHz power) for varying times (5, 10, and 15 min) at 25 °C, and their combination with different packaging types, i.e., vacuum and passive modified atmosphere packaging (passive MAP), on the physical, chemical, and sensory properties of pomegranate arils. The results showed that the combined treatment of ultrasonic and vacuum packaging leads to improving pomegranate arils' quality and shelf life. The treatments significantly reduced weight loss (30%) and decay (16%) compared to the control. At the end of the storage time, the lowest bacterial count (0.25 log CFU/g) and the lowest yeast and mold count (0.37 log CFU/g) were obtained in ultrasound-treated arils for 15 min that were vacuum packed. In addition, they preserved the total soluble solids, titratable acidity, antioxidant capacity, phenolic compounds, and anthocyanin, leading to improving the organoleptic properties of arils. However, in terms of taste and overall quality, greater scores were recorded by panelists in vacuum-packed arils than in passive MAP. Overall, arils that were treated with ultrasound for 10 min and then vacuum packed had the best results compared to the control and other combination treatments.

**Keywords:** minimally processed; passive MAP; postharvest; shelf life; vacuum





## 1. Introduction

Pomegranate is a valuable edible and medicinal fruit with high economic value. Pomegranate is an important fruit of the subtropical regions that grows in the Mediterranean climate that has nutritional value due to the presence of antioxidants, alkaloids, phenolic compounds, and vitamins [1]. The medicinal characteristics of pomegranate have been discussed, highlighting the importance of ellagic acid and flavones [2], and their effectiveness has been confirmed in the treatment of prostate cancer [3]. In addition, pomegranate is heart-strengthening, removes toxins and heat from the body and cleans the stomach [4].

Currently, due to lifestyle changes and an awareness of the health benefits of fruits, consumer demand for ready-to-eat fruits has increased [5]. Therefore, minimally processed pomegranate can be a suitable product to increase the consumption of this valuable fruit. Minimally processed and packaged fruit is a cost-effective way to ensure that 100% of the product is usable. However, the physical damage caused by cutting and peeling increases the respiration rate and ethylene production [6]. Peeling leads to the leakage of nutrients, acceleration of enzymatic and microbial reactions, color change, texture change and weight loss, which ultimately reduces the quality of the product [7]. One of the most important factors limiting the overall quality and shelf life of pomegranate arils is microbial growth,

as well as the loss of nutrition caused by active metabolic processes related to enzyme activity, respiration, or oxidation [8]. Therefore, maintaining the quality and microbial safety of pomegranate arils is a challenge and requires suitable strategies to preserve the quality and organoleptic characteristics of the fruit while extending the shelf life.

Various postharvest techniques, such as modified atmosphere packaging (MAP) and non-destructive ultrasound (US) method are considered by many researchers. Functional packagings, such as modified atmosphere (MA) and vacuum (V), have been considered for maintaining fresh products [9]. Generally, there are two types of MAP in food industry, active and passive. To make an active MAP, a common method is the injection of a determined respiratory gas (oxygen and carbon dioxide) composition, which is different from an air atmosphere, into the package. While in passive MAP, the product is sealed in the package content air. MAP of minimally processed pomegranate arils offers additional tools for optimal use and value addition [10]. Low-temperature conditions, combined with low oxygen levels and moderate to high carbon dioxide levels, have been used to extend the shelf life of fresh-cut fruits [11,12], and have led to a decrease in respiration and ethylene production, inhibiting or delaying enzymatic reactions [13]. A low-oxygen atmosphere can potentially reduce chilling injury, weight loss, and delayed ripening [14]. Packaging can be significantly effective in increasing the phytochemical compounds of fruits, and delaying the growth of aerobic spoilage microorganisms [15].

Vacuum packaging (VP) involves storing a product in an entirely impermeable package that has been evacuated from the air. Limiting access to oxygen reduces respiration [16]. Additionally, reducing the temperature and pressure has a positive effect during storage [17]. Passive MAP [18,19], active MAP [20], and vacuum packaging [21] extended the postharvest shelf life of the whole pomegranate fruit and arils cv. Shishe-Kab. In another study, the passive MAP of pomegranate arils cv. Wonderful effectively preserved the taste, aroma, and overall acceptability [22].

US is considered a modern technology in the food industry. Ultrasonication has been approved by the U.S. Food and Drug Administration (FDA) as an alternative heat treatment technology for food processing and analysis. US is a new non-thermal technology that reduces the microbial load and increases the nutritional value by inhibiting the loss of bioactive compounds. The mechanical, chemical and biochemical effects created in food products depend on the intensity of the US waves [23]. Besides, the inactivation of microorganisms and enzymes in food products can be achieved by emission of ultrasound waves (alone or combined with pressure and/or heat) [24,25]. For an instant, US treatment (40 kHz; 350 W; for 10 min) has been shown to inhibit the activity of polygalacturonase and pectate lyase enzymes, thus delaying ripening and extending the shelf life of jujube fruit [26]. Similarly, ultrasonic treatment (50 kHz; for 5 min) has been found reduce ethylene synthesis and soluble solids, while maintaining the firmness of apple fruit [27]. The synergistic effect of the US and other postharvest treatments has been reported to maintain the quality and increase the shelf life of fresh-cut and whole fruits [28,29]. A recent study regarding the effects of US (40 kHz; for 15 and 30 min) on pomegranate aril cv. Rabab-e-Neyriz has demonstrated that it can significantly prevent anthocyanin and ascorbic acid degradation [30]. Moreover, the highest amount of anthocyanin, ascorbic acid, and antioxidant activity was found in arils treated with US for 15 min, and the arils treated with US for 30 min showed no signs of decay compared to the control group. However, little information is available regarding the combined effect of US and packaging treatments on minimally processed pomegranate aril. Therefore, this research aimed to study the combined effect of US and MAP on the qualitative characteristics and shelf life of pomegranate arils cv. Shishe-Kab.

## 2. Materials and Methods

To perform the experiment, pomegranate fruit cv. Shishe-Kab was used, which has the largest area under cultivation among pomegranate cultivars in South Khorasan province, Iran. The fruits were harvested at the commercial maturity stage (average weight $250 \pm 30$ g)

from a commercial orchard. After being transferred to the horticultural laboratory of the University of Birjand, uniform fruits that were free of damage and infection were selected and immediately washed with distilled water. After surface drying, the fruits cut vertically, and the arils were manually collected. After that, pomegranate arils were treated with the US (SONOREX SUPER RK 510H model, BANDELIN electronic GmbH & Co. KG, Berlin, Germany) at 35 kHz power, for varying times (5, 10 and 15 min) at 25 °C. The arils were then packed in polyethylene films with two types of atmospheres, vacuum (VP) and passive MAP, with 100 g of arils in each experimental unit. The packages were stored at 5 °C and 85% RH, and sampling was carried out every seven days to evaluate the traits.

### 2.1. Weight Loss

To measure the weight loss, the packages were weighed before storage, and at the end of each stage of storage (0, 7 and 14 days). The weight loss of the packages was calculated using Equation (1):

$$WL\% = [(W_i - W_S)/Wi] \tag{1}$$

$W_i$ and $W_s$ are the initial weight and secondary weight, respectively.

### 2.2. Decay Rate and Microbial Load

The decay of arils is determined through visual observations during storage. Therefore, after the appearance of symptoms such as browning and mold growth, the number of infected arils were counted and the percentage of decay was calculated using Equation (2):

$$Decay\ rate\ \% = (N_i/N_t) \times 100 \tag{2}$$

$N_i$ and $N_t$ are No. of infected arils and No. of total arils, respectively.

The colony counting method was used to investigate aril microbial populations [31]. Total bacteria and total fungi and yeast were estimated on the plate count agar (PCA) incubated at 37 °C for two days, and on potato dextrose agar (PDA) incubated at 28 °C for three days, respectively.

### 2.3. Color Characteristics (L, a*, b*)

The color characteristics were measured using a colorimeter (TES 135— SAEN Co, Taipei, Taiwan). The results were expressed based on Hunter's color characteristics, where *L* represents lightness 0 to 100; therefore, 100 is the maximum lightness and 0 is darkness. Index *a\** indicates red to green color, and positive and negative values indicate an increase in red or green color, respectively. In other words, positive values of *a\** represent red coloration, while negative values of *a\** correspond to green coloration. Index *b\** indicates yellowness to blue color, and positive and negative values indicate the increase in yellow or blue color, respectively [32]. Similarly, positive values of *b\** represent yellow coloration, while negative values of *b\** correspond to blue coloration.

### 2.4. Total Soluble Solids (TSS)

A handheld refractometer (RF10, 0–32° Brix, Extech Co., Chicago, IL, USA) was used to measure total soluble solids at room temperature, and the data were expressed in the degree Brix.

### 2.5. pH and Titratable Acidity (TA)

The pH of fruit juice was measured with a digital pH meter (PP-203 Model, EZDO Gondo Electronic Co., Ltd., Taipei Nan Kang, Taiwan). To measure TA, 2–3 drops of phenolphthalein as an indicator were added to 5–10 cc of fruit juice and titrated with a 0.1 N NaOH solution to appear as a light pink color [33]. TA is calculated according to Equation (3) and expressed as a percentage.

$$TA\% = (N \times V_1 \times Eq_{wt}/V_2 \times 1000) \times 100 \tag{3}$$

$V_1$ and $V_2$ are the volume of used NaOH and sample volume, respectively, based on the weight equivalent of the predominant acid (citric acid = 70 g).

### 2.6. Total Phenols Content (TPC)

Two gr of aril was ground in liquid nitrogen and then 10 mL of ethanol (96% *v/v*) was added to the extract. After centrifugation at 4500 rpm for 15 min at 4 °C [34], supernatant was used to measure total phenol using Folin–Ciocalteu method. Then, 0.5 mL of the ethanolic extract (supernatant phase) was transferred to the test tubes and after incubation at room temperature for 5 min was added 0.5 mL of Folin–Ciocalteu reagent under the same conditions. Then, 2 mL of sodium bicarbonate (200 g $L^{-1}$) was added and shaken slowly. After incubation at room temperature for 15 min, 10 mL of deionized water added and centrifuged at 4000 rpm for 5 min at 4 °C. The absorbance of the samples measured at 750 nm by the spectrophotometric method. The concentration of TPC is expressed as mg of gallic acid per 100 gr of fresh weight [35]. The evaluation was based on the standard curve of gallic acid.

### 2.7. Total Anthocyanin Content (TAC)

The anthocyanin concentration was determined by the pH differential method. Briefly, the aril sample extract was mixed with a potassium chloride buffer (pH 1.0, 0.025 M) and sodium acetate buffer (pH 4.5, 0.4 M), separately. The absorbance was measured at 510 and 700 nm [36]. The absorbance difference between different buffers was calculated using Equation (4):

$$A = (A510 - A700)pH1.0 - (A510 - A700)pH4.5 \tag{4}$$

TAC based on the concentration of cyanidin-3-glucoside was calculated using Equation (5):

$$TAC(mgL^{-1}) = A \times MW \times DF \times 100/f \times d \tag{5}$$

where A is the absorbance, MW is the molecular weight of cyanidin-3-glucoside (433.3), DF is the dilution factor (10) and f is the molar absorptive coefficient of cyanidin-3-glucoside (15,600).

### 2.8. Antioxidant Activity (AA)

As mentioned earlier, two gr of aril was ground in liquid nitrogen and then 10 mL of ethanol (96% *v/v*) was added to the extract. The supernatant phase was discarded and extraction was performed by adding 10 mL of ethanol (70% *v/v*) to the sediments (repeated twice) [34]. The extract was centrifuged at 4500 rpm for 15 min and the supernatant phase was used to determine the antioxidant activity using DPPH (2,2-diphenyl-1-pickrylhydrazyl) free radicals according to the method of De Ancos et al. [37] with some modifications. Therefore, 0.5 mL of ethanol extract was added to 2 mL of DPPH free radical ethanol solution (0.25 mM). After 2 h of incubation in the darkness, the absorbance was measured at 517 nm using a spectrophotometer. The results are expressed as DPPH radical inhibition percentage using Equation (6):

$$DPPH\ radical\ scavenging\ activity\ \% = (1 - Abs_{sample}/Abs_{control}) \times 100 \tag{6}$$

where $Abs_{sample}$ and $Abs_{control}$ represent the absorbance of the sample and control, respectively.

### 2.9. Organoleptic Characteristics

The sensory quality of pomegranate arils was evaluated by eight untrained panelists through a 5-point hedonic scale. Therefore, the quality score of 5 is considered desirable for arils. A score of 3 is considered a threshold limit, and a quality score higher than 3 has an acceptable quality the consumer. A quality score of 1 is considered unacceptable for the consumer [38].

*2.10. Statistical Analysis*

The experiment was conducted as a factorial design based on a completely randomized design (CRD) with seven treatments and four replications. The factors included US time (5, 10 and 15 min), type of packaging (vacuum and passive MAP), and storage time (1, 7, and 14 days). Data analysis was performed using the statistical software GenStat (Discovery Edition, Version 9.2, 2007, VSN. International Ltd., Hemel Hempstead, UK), and the comparison of average data was performed using Fisher's least significant difference (LSD) test at the 5% probability level.

**3. Result and Discussion**

The effects of different US treatments, storage time, and their interaction effects on the all evaluated traits of arils were significant ($p < 0.01$). However, in terms of L value, storage time, and its interaction with US treatment was not significant. Regarding pH, US treatment and its interaction with storage time was not significant.

*3.1. Weight Loss*

The effect of different treatments indicated that the minimum weight loss (1.36%) was achieved in 15 min US-VP treatment. However, it was not significantly ($p < 0.01$) different from 5 min US-VP (1.39%) and 10 min US-VP (1.47%) (Table 1). The effect of storage time showed that the minimum weight loss (zero) was obtained on the first day with a significant ($p < 0.01$) difference compared with the other days (Table 1). The interaction effects of different treatments and storage time showed that the minimum weight loss (zero) was obtained in all the treatments on the first day without a significant ($p < 0.01$) difference between them (Figure 1a).

**Table 1.** Effects of different US pretreatments and packaging type, vacuum packaging (VP) or passive MAP, and storage time on the weight loss and decay rate of pomegranate arils cv. Shishe-Kab.

| Pretreatments | Weight Loss (%) | Decay Rate (%) |
|---|---|---|
| Control (without US and packaging) | 2.02 ± 0.3 [a] | 24.00 ± 2.0 [a] |
| 5 min US—VP | 1.39 ± 0.2 [c] | 12.11 ± 1.7 [cd] |
| 5 min US—Passive MAP | 1.78 ± 0.2 [ab] | 14.33 ± 1.2 [bc] |
| 10 min US—VP | 1.47 ± 0.1 [bc] | 10.77 ± 0.8 [de] |
| 10 min US—Passive MAP | 1.80 ± 0.3 [a] | 14.11 ± 1.4 [bc] |
| 15 min US—VP | 1.36 ± 0.2 [c] | 8.11 ± 0.7 [e] |
| 15 min US—Passive MAP | 1.81 ± 0.3 [a] | 15.55 ± 1.3 [b] |
| Storage time | | |
| First day | 0.0 [c] | 0.0 [c] |
| 7th day | 1.67 ± 0.2 [b] | 2.04 ± 0.3 [b] |
| 14th day | 3.32 ± 0.4 [a] | 40.38 ± 4.6 [a] |

Means followed by similar letters are not significantly different according to least significant difference (LSD) test ($p \leq 0.05$).

The weight loss of pomegranate aril had an increasing trend during storage. In accordance with our results, increasing the US time for the juice of the Plum Gold Drops has led to the maintenance of fresh weight [39]. The arils of VP had the lowest weight loss, which may be related to gaseous composition and their effect on metabolic processes. Similar to our findings, MAP decreased the weight loss of pomegranate in the Shishe-Kab [40] and the Wonderful cultivars [41,42].

*3.2. Decay Rate and Microbial Load*

The effect of different treatments showed that the minimum decay (8.11%) was observed in 15 min US-VP. However, it was not significantly ($p < 0.01$) different from 10 min US-VP (10.77%) (Table 1). The effect of storage time showed that the minimum decay (zero) was observed on the first day with a significant ($p < 0.01$) difference compared with the

other days (Table 1). The interaction effects of different treatments and storage time showed that the minimum decay (zero) was observed in all the treatments on the first and seventh day (except for the control on the seventh day) without a significant ($p < 0.01$) difference between them (Figure 1b). The results of microbial analysis also indicated that the US and packaging significantly decreased the growth rate of bacteria, fungi and yeast on arils in all combination treatments compared to the control. On day 0, bacterial, yeast and mold counts were below the detection limit (<1 log CFU/g) in all treatments. However, at the end of storage time, the lowest bacterial count (0.25 log CFU/g) was observed in US-VP for 15 min, whereas the highest (1.38 log CFU/g) was found in the control sample. Similarly, the lowest yeast and mold count (0.37 log CFU/g) was obtained in the US-VP for 15 min, and the highest (1.61 log CFU/g) was observed in control fruit.

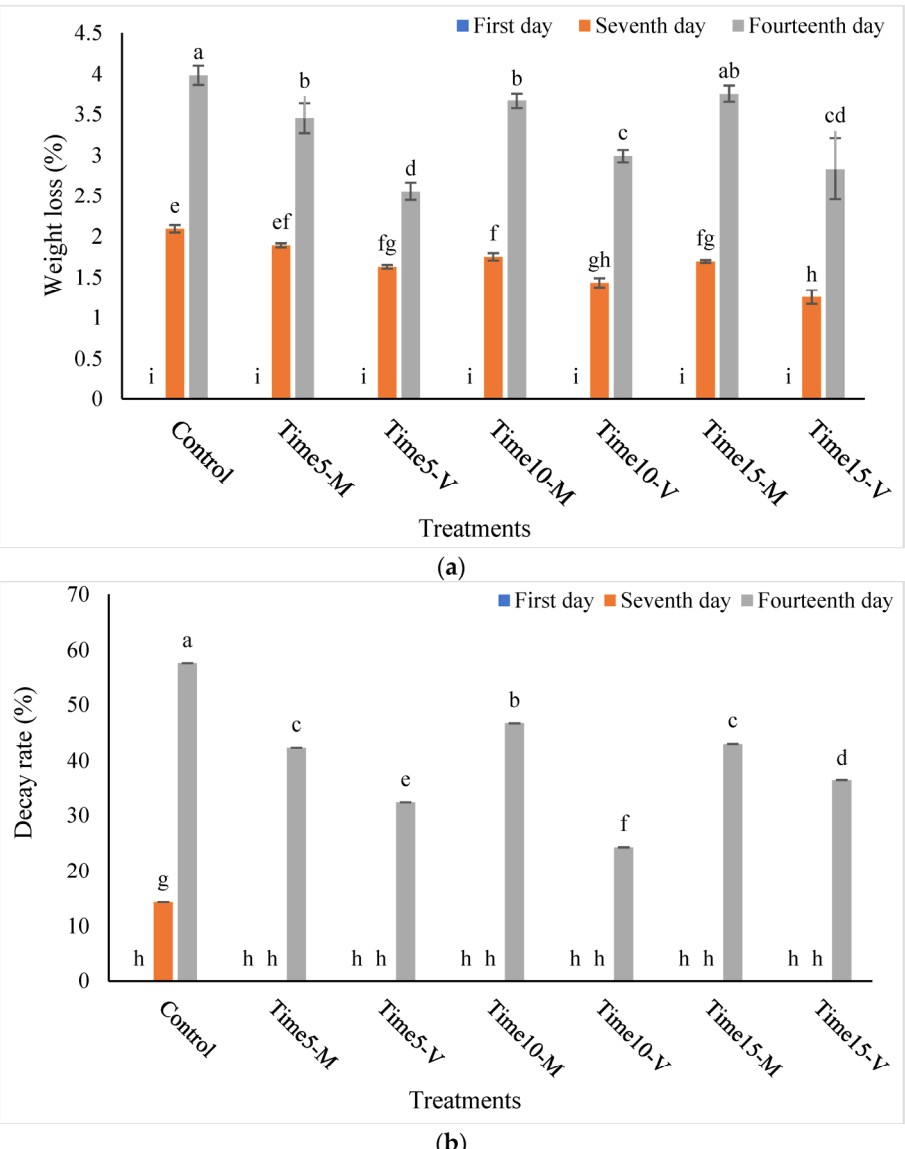

**Figure 1.** Interaction effects of different ultrasound (US) pretreatments and storage time on weight loss (**a**) and decay rate (**b**) of arils stored at 5 °C for 14 days. Means followed by similar letters are not significantly different according to least significant difference (LSD) test ($p \leq 0.05$). V: vacuum packaging, M: passive MAP.

In line with our observations, US treatment alone and in combination with peracetic acid reduced the rot of Japanese parsnip compared to the control. However, the combined

treatment was more effective [43]. Natural compounds prevent decay and aging by inducing systemic resistance (ISR), increasing antioxidants, and reducing respiration and ethylene production [44]. Short-term controlled atmosphere treatments have been proven to reduce weight loss and decay [45], which is related to cell wall strength and the expression of plant defense genes in response to the modified atmosphere [46]. Rokalla et al. [20] also indicated that MAP significantly reduced microorganisms in pomegranate aril packages stored for five days at 25 °C compared to the control. The other researchers stated that the effect of the modified atmosphere in the decay control is related to the delay of ripening and the reduction of respiration [47].

### 3.3. Color Characteristics (L, a*, b*)

The effect of different treatments showed that the maximum $L$ value (7.28) was observed in 5 min US-VP. However, it was not significantly ($p < 0.01$) different from other treatments (except 10 min US–passive MAP) (Table 2).

**Table 2.** Effects of different US pretreatments and packaging type, vacuum packaging (VP) or passive MAP, and storage time on color parameters ($L$, $a*$, and $b*$ index) of pomegranate arils cv. Shishe-Kab.

| Pretreatments | $L$ Value | $a*$ Index | $b*$ Index |
|---|---|---|---|
| Control (without US and packaging) | 7.04 ± 0.4 [ab] | 19.80 ± 1.6 [c] | 1.50 ± 0.09 [b] |
| 5 min US-VP | 7.28 ± 0.5 [a] | 22.35 ± 1.9 [a] | 1.59 ± 0.07 [ab] |
| 5 min US–Passive MAP | 6.15 ± 0.4 [ab] | 20.88 ± 1.4 [bc] | 1.56 ± 0.08 [ab] |
| 10 min US-VP | 6.75 ± 0.6 [ab] | 21.62 ± 1.7 [ab] | 1.64 ± 0.06 [a] |
| 10 min US–Passive MAP | 6.53 ± 0.5 [ab] | 20.94 ± 1.8 [bc] | 1.57 ± 0.09 [ab] |
| 15 min US-VP | 6.97 ± 0.7 [ab] | 22.47 ± 2.1 [a] | 1.60 ± 0.05 [ab] |
| 15 min US–Passive MAP | 5.74 ± 0.8 [b] | 21.27 ± 1.8 [ab] | 1.48 ± 0.07 [b] |
| Storage time | | | |
| 1st day | - | 23.13 ± 1.7 [a] | 1.82 ± 0.08 [a] |
| 7th day | - | 21.18 ± 1.9 [b] | 1.51 ± 0.07 [b] |
| 14th day | - | 19.69 ± 2.2 [c] | 1.35 ± 0.06 [c] |

Means followed by similar letters are not significantly different according to least significant difference (LSD) test ($p \leq 0.05$).

The effect of different treatments showed that the maximum $a*$ index was observed in 15 and 5 min US-VP with values of 22.47 and 22.35, respectively. However, it was not significantly ($p < 0.01$) different from 10 min US-VP (21.62%) (Table 2). The effect of storage time showed that the maximum $a*$ index (23.13) was observed on the first day with a significant ($p < 0.01$) difference compared with the other days (Table 2). The interaction effects of different treatments and storage time showed that the maximum $a*$ index observed in all the treatments on the first day without a significant ($p < 0.01$) difference between them (Figure 2).

The effect of different treatments showed that the maximum $b*$ index observed in 10 min US-VP (1.64); however, it was not significantly ($p < 0.01$) different from 5 and 15 min US-VP, and 5 and 10 min US–passive MAP (Table 2). The effect of storage time indicated that the maximum $b*$ index (1.82) was obtained on the first day of storage with a significant ($p < 0.01$) difference compared with the other days (Table 2).

Consistent with our results, the power, temperature and time of the US had a significant effect on the color characteristics of strawberry extract [48]. They found that US treatments with higher power, temperature and for a longer time significantly decreased the values of color parameters ($L$, $a*$, and $b*$) and total anthocyanin content of strawberry juice compared to the untreated sample. Also, the modified atmosphere had a significant effect on the $L$ value of pomegranate arils [49], and maintained the lightness of the aril during storage [50]. However, little changes in $L$ value in all treatments were observed. The reduction in the intensity of the red color ($a*$ index) is related to anthocyanin. In line with our observations in terms of the effect of VP and passive MAP on $a*$, the MAP of

pomegranate arils had a significant effect on the *a** index and increased it by maintaining the anthocyanin content [51].

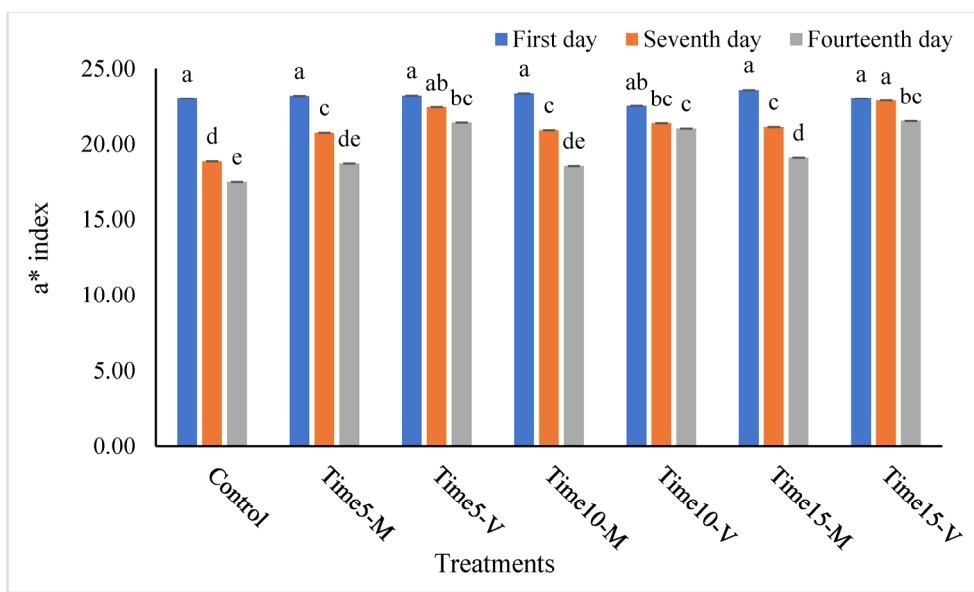

**Figure 2.** Interaction effects of different ultrasound (US) pretreatments and storage time on *a** index of arils stored at 5 °C for 14 days. Means followed by similar letters are not significantly different according to least significant difference (LSD) test ($p \leq 0.05$). V: vacuum packaging, M: passive MAP.

*3.4. Total Soluble Solids (TSS)*

The effect of different treatments showed that the highest TSS was found in 10 and 5 min US-VP with the amount of 16.01% and 15.83%, respectively, which had a significant ($p < 0.01$) difference compared with the other treatments (Table 3). The effect of storage time indicated that the highest TSS (15.86%) was observed on the seventh day with a significant ($p < 0.01$) difference compared with the other days (Table 3). The interaction effects of different treatments and storage time showed that the highest TSS was found in the control on the seventh day, 5 min US–passive MAP on the 7th day, 5 min US-VP on the 14th day, 10 min US-VP on the 7th and 14th day (Figure 3a).

**Table 3.** Effects of different US pretreatments and packaging type, vacuum packaging (VP) or passive MAP, and storage time on total soluble solids (TSS), titratable acidity (TA), and pH of pomegranate arils cv. Shishe-Kab.

| Pretreatments | TSS (%) | TA (%) | pH |
|---|---|---|---|
| Control (without US and packaging) | 15.30 ± 1.7 [c] | 1.61 ± 0.08 [b] | - |
| 5 min US-VP | 15.83 ± 1.8 [ab] | 2.11 ± 0.10 [a] | - |
| 5 min US–Passive MAP | 15.31 ± 1.7 [c] | 2.01 ± 0.09 [a] | - |
| 10 min US-VP | 16.01 ± 1.5 [a] | 2.10 ± 0.10 [a] | - |
| 10 min US–Passive MAP | 15.35 ± 1.9 [c] | 2.03 ± 0.08 [a] | - |
| 15 min US-VP | 15.50 ± 1.6 [bc] | 2.12 ± 0.07 [a] | - |
| 15 min US–Passive MAP | 15.37 ± 2.1 [c] | 2.00 ± 0.11 [a] | - |
| Storage time | | | |
| 1st day | 15.35 ± 1.5 [b] | 2.34 ± 0.09 [a] | 3.65 ± 0.2 [a] |
| 7th day | 15.86 ± 1.7 [a] | 1.96 ± 0.07 [b] | 3.33 ± 0.3 [b] |
| 14th day | 15.36 ± 1.3 [b] | 1.68 ± 0.07 [c] | 3.07 ± 0.4 [c] |

Means followed by similar letters are not significantly different according to least significant difference (LSD) test ($p \leq 0.05$).

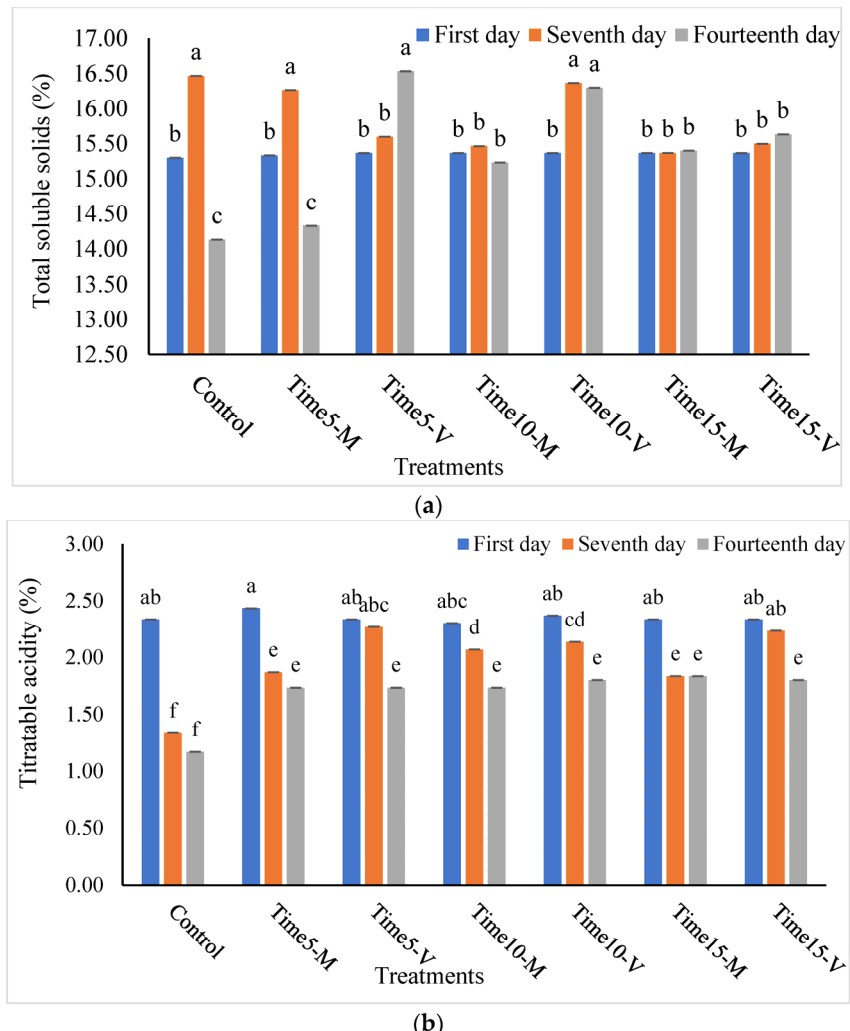

**Figure 3.** Interaction effects of different ultrasound (US) pretreatments and storage time on total soluble solids (**a**) and titratable acidity (**b**) of arils stored at 5 °C for 14 days. Means followed by similar letters are not significantly different according to least significant difference (LSD) test ($p \leq 0.05$). V: vacuum packaging, M: passive MAP.

In the present study, ultrasonic and VP prevented the decrease in soluble solids. On contrary, US had no significant effect on the TSS of pomegranates juice [52]. This inconsistency is likely due to different pomegranate cultivars, and/or various US power, time and temperature applied. Although the soluble solids value in the modified atmosphere was higher than in the normal atmosphere [53]. It seems that the modified atmosphere reduces the fruit metabolism and respiration rate and provides more controlled conditions for the utilizing of vacuolar sugars [54]. Shishe-Kab pomegranate cultivar, as a non-climacteric fruit, is harvested with roughly 15% TSS, so generally few changes in TSS values occurred during storage time.

### 3.5. pH and Titratable Acidity (TA)

The effect of storage time showed that the highest pH (3.65) was found on the first day with a significant ($p < 0.01$) difference compared with the other days (Table 3).

The effect of different treatments showed that the highest TA (2.12%) was found in 15 min US-VP. However, it was not significantly ($p < 0.01$) different from other treatments (except control) (Table 3). The effect of the storage time showed that the highest TA (2.43%) was found on the first day with a significant ($p < 0.01$) difference compared with the other days (Table 3). The interaction effects of different treatments and storage time indicated

that the highest TA found in 5 min US—passive MAP. However, it was not significantly ($p < 0.01$) different from other treatments at the same time (Figure 3b).

Similar to our observations, the pH of pomegranate arils decreased during storage, although the changes were non-significant [49]. The reduction of TA during storage is related to metabolic activities and acid consumption in respiration [55]. In this study, the preservation of TA arils treated with US indicates maintaining cell structure and less consumption of organic acids. Similarly, the application of the US reduced the degradation of organic acids in strawberries [56]. In the current research, VP maintained the TA of arils, likely due to the partial oxygen pressure that reduces destructive metabolisms and maintains titratable acidity [57].

### 3.6. Total Phenols Content (TPC)

The effect of different treatments showed that the highest TPC (128 mg 100 g$^{-1}$ F.W.) was obtained in 10 min US-VP. However, it was not significantly ($p < 0.01$) different from other treatments (except control) (Table 4). The effect of the storage time showed that the highest TPC (137.41 mg 100 g$^{-1}$ F.W.) was found on the first day with a significant ($p < 0.01$) difference compared with the other days (Table 4). The interaction effects of different treatments and storage time indicated that the highest TPC was recorded in all the treatments on the first day without a significant ($p < 0.01$) difference between them (Figure 4a).

**Table 4.** Effects of different US pretreatments and packaging type, vacuum packaging (VP) or passive MAP, and storage time on total phenol content (TPC), total anthocyanin content (TAC), antioxidant activity (AA), and taste index of pomegranate arils cv. Shishe-Kab.

| Pretreatments | TPC (mg 100 g$^{-1}$ F.W.) | TAC (mg 100 g$^{-1}$ F.W.) | AA (%) | Taste Index |
|---|---|---|---|---|
| Control (without US and packaging) | 110.98 ± 6.7 [b] | 140.00 ± 11.6 [c] | 60.99 ± 5.4 [d] | 3.00 ± 0.4 [c] |
| 5 min US-VP | 127.78 ± 8.7 [a] | 148.01 ± 12.7 [a] | 63.44 ± 5.7 [a] | 4.11 ± 0.4 [ab] |
| 5 min US–Passive MAP | 127.19 ± 7.2 [a] | 141.29 ± 10.7 [bc] | 61.64 ± 6.7 [cd] | 3.88 ± 0.5 [ab] |
| 10 min US-VP | 128.00 ± 9.4 [a] | 146.60 ± 9.7 [a] | 63.95 ± 7.1 [a] | 4.33 ± 0.6 [a] |
| 10 min US–Passive MAP | 127.19 ± 6.5 [a] | 139.78 ± 11.2 [c] | 62.36 ± 5.8 [b] | 3.66 ± 0.3 [b] |
| 15 min US-VP | 127.76 ± 10.2 [a] | 142.80 ± 10.7 [b] | 64.02 ± 7.4 [a] | 4.11 ± 0.5 [ab] |
| 15 min US–Passive MAP | 127.30 ± 9.1 [a] | 142.83 ± 12.8 [b] | 62.21 ± 4.9 [bc] | 3.55 ± 0.7 [bc] |
| Storage time | | | | |
| 1st day | 137.41 ± 10.5 [a] | 151.33 ± 11.5 [a] | 65.16 ± 7.7 [b] | 5.00 ± 0.6 [a] |
| 7th day | 129.56 ± 8.7 [b] | 150.81 ± 8.7 [a] | 70.39 ± 6.8 [a] | 4.00 ± 0.4 [b] |
| 14th day | 108.55 ± 9.3 [c] | 126.99 ± 10.1 [b] | 52.43 ± 6.4 [c] | 2.42 ± 0.3 [c] |

Means followed by similar letters are not significantly different according to least significant difference (LSD) test ($p \leq 0.05$).

Consistent with our results, increasing the phenolic compounds of blackberry [58], grapefruit [59], and apple extracts [60] had observed after the processing of low-intensity US, due to the better extraction of phenols from suspended particles [61]. In line with our results, a decrease in the total phenolic content of pomegranate (cv. Wonderful) during storage has been reported [41]. In the present study, the total phenol of arils was significantly higher in the vacuum and passive MAP, likely due to increasing the phenylpropanoid synthesis pathway under low oxygen conditions [8]. On the other hand, the oxidation of phenols via polyphenol oxidase enzyme is in the presence of oxygen. Therefore, reducing access to oxygen can preserve phenolic compounds [62]. Moreover, the modified atmosphere helps to preserve the polyphenolic compounds via reducing respiration in blue honeysuckle fruit [63].

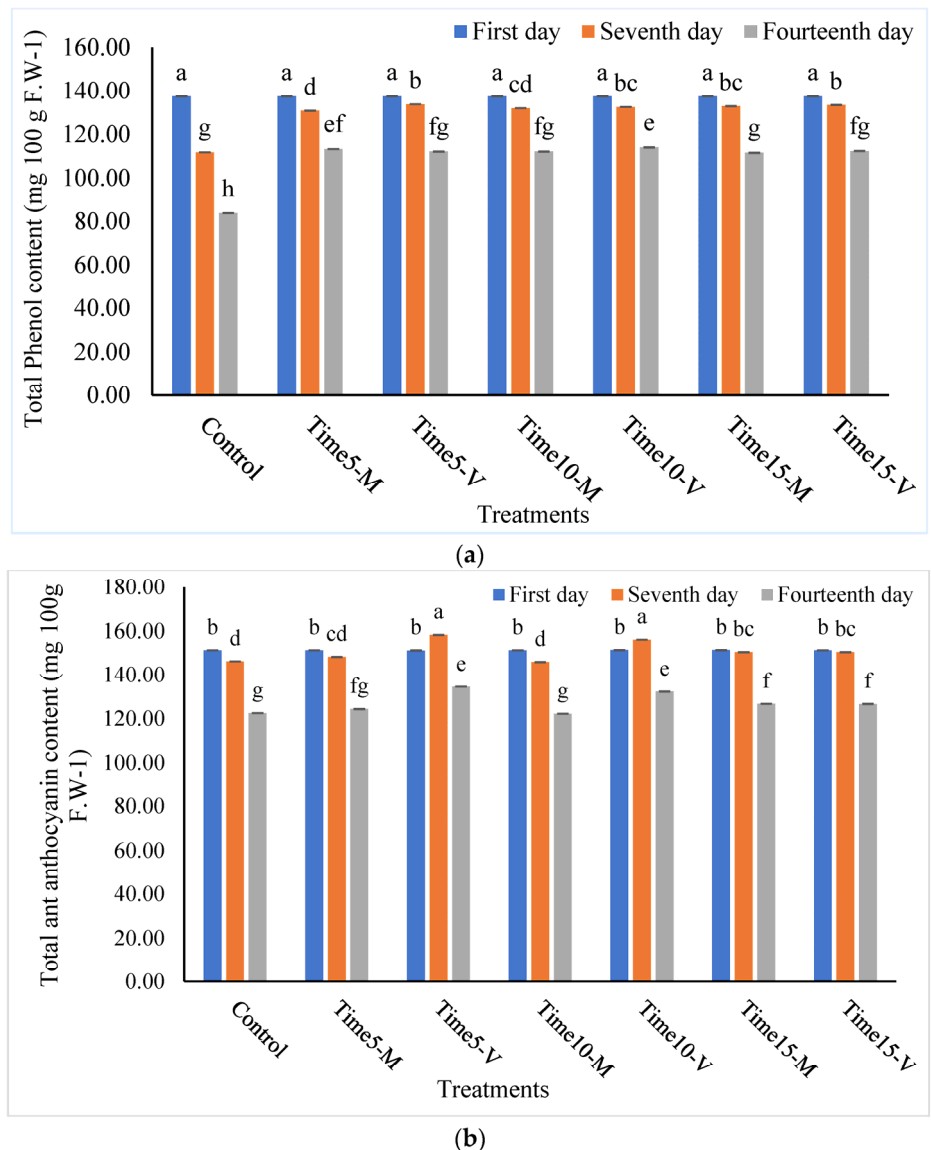

**Figure 4.** Interaction effects of different ultrasound (US) pretreatments and storage time on total phenol content (**a**) and total anthocyanin content (**b**) of arils stored at 5 °C for 14 days. Means followed by similar letters are not significantly different according to least significant difference (LSD) test ($p \leq 0.05$). V: vacuum packaging, M: passive MAP.

*3.7. Total Anthocyanin Content (TAC)*

The effect of different treatments showed that the highest TAC was found in 5 and 10 min US-VP with the amount of 148.01 and 146.60 mg 100 g$^{-1}$ F.W., respectively. Also, it had a significant ($p < 0.01$) difference from other treatments (Table 4). The effect of storage time showed that the highest TAC was recorded on the first and seventh day, with the amount of 151.33 and 150.81 mg 100 g$^{-1}$ F.W., respectively, with a significant ($p < 0.01$) difference compared to the fourteenth day (Table 4). The interaction effects of different treatments and storage time showed that the highest TAC was found in 10 and 15 min US-VP treatment on the seventh day (Figure 4b).

In the extraction of anthocyanins, high temperatures and long extraction times degrade the compounds. An alternative extraction method is required to overcome these problems. Among the alternative extraction techniques, ultrasound-assisted extraction (UAE) is considered environmentally friendly and recognized for its short extraction time and reduced consumption of solvents [64,65]. Consistent with our results, the US with low

intensity and short time (UAE) caused a slight increase in anthocyanins in red grape extract due to the better extraction of anthocyanins from suspended particles [65,66]. Similarly, packaging of the pomegranate arils (cv. Wonderful) in passive MAP [22] and Lonicera caerulea in a low oxygen atmosphere [63] preserved anthocyanin content.

### 3.8. Antioxidant Activity (AA)

The effect of different treatments showed that the highest AA was found in 15, 10 and 5 min US-VP with the amount of 64.02, 63.95, and 63.44%, respectively, with a significant ($p < 0.01$) difference from other treatments (Table 4). The effect of storage time showed that the highest AA (70.39%) was observed on the seventh day with a significant ($p < 0.01$) difference compared with the other days (Table 4). The interaction effects of different treatments and storage time showed that the highest AA was found in 15 and 10 min US-VP on the seventh day (Figure 5a).

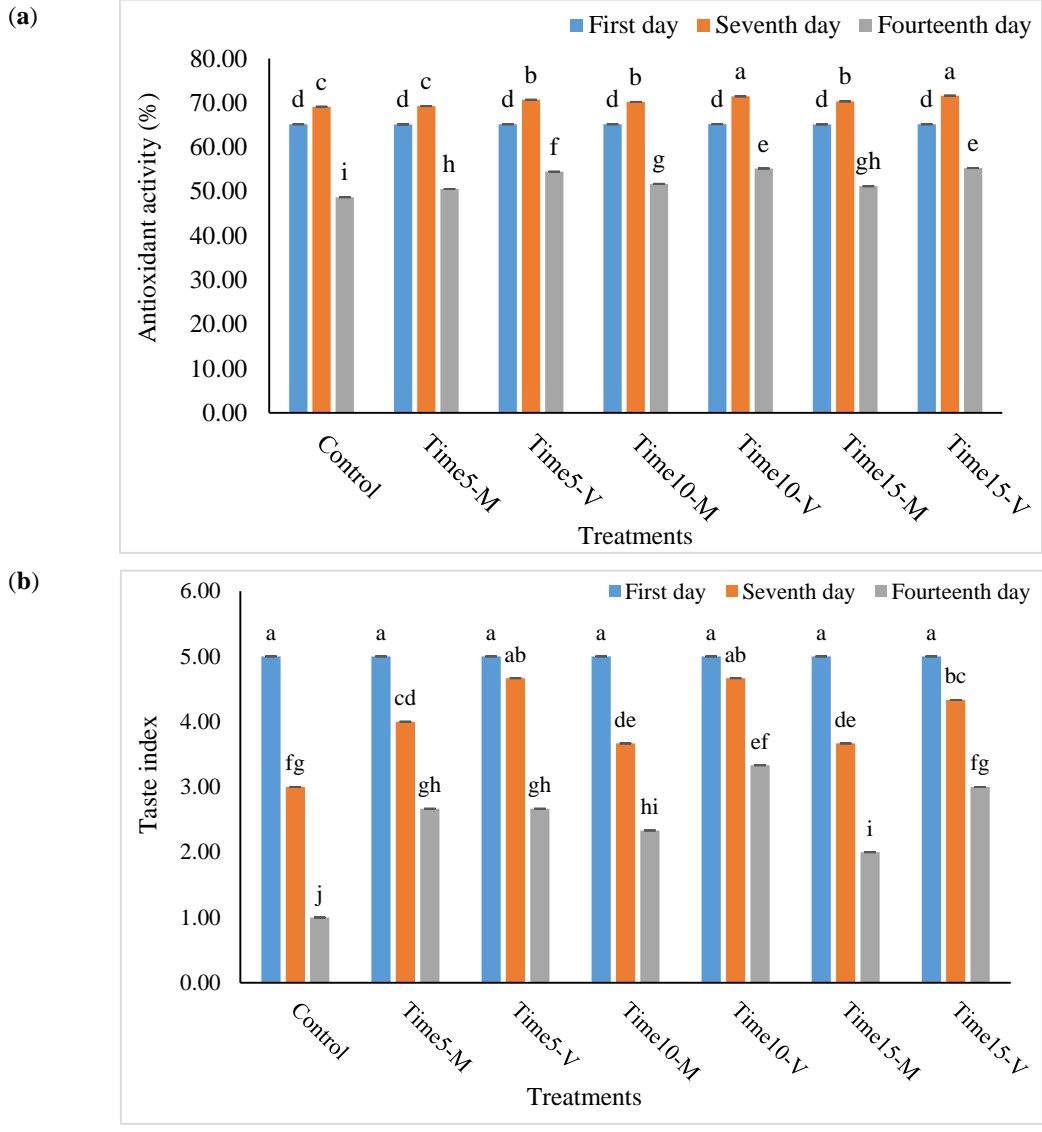

**Figure 5.** Interaction effects of different ultrasound (US) pretreatments and storage time on antioxidant activity (**a**) and taste index (**b**) of arils stored at 5 °C for 14 days. Means followed by similar letters are not significantly different according to least significant difference (LSD) test ($p \leq 0.05$). V: vacuum packaging, M: passive MAP.

In line with our results, the US treatment led to the maintenance of phenolic compounds and improved antioxidant capacity [61,67,68]. The increasing phenolic compounds and antioxidant properties are due to hydroxylation in the positions of ortho- (o-), meta- (m-), and para- (p-) monophenols. Increasing the US time, the juice *Citrus reticulata* cv. Bakraei significantly increased ABTS and DPPH inhibition [69]. US reduces the loss of nutrients and therefore preserves antioxidant activity [70]. Ultrasonication, which is the application of US at low temperature, as an abiotic stress, could promote the synthesis of bioactive compounds, anthocyanins, antioxidants and ascorbic acid via the simulation of their physiological activities [30].

### 3.9. Organoleptic Characteristics

The highest taste index score (4.33) was obtained in 10 min US-VP. However, it was not significantly ($p < 0.01$) different from 5 and 15 min US-VP (4.11) and 10 min US–passive MAP (3.88) (Table 4). The effect of storage time showed that a high score (5) was obtained on the first day with a significant ($p < 0.01$) difference compared with the other days (Table 4). The interaction effects of different treatments and storage time showed that the high score was obtained in all the treatments on the first day without a significant ($p < 0.01$) difference between them (Figure 5b).

Our results indicate that the combination of US and VP preserving the taste of arils at the desired level. The effectiveness of the modified atmosphere in maintaining freshness is due to the creation of saturated relative humidity and the reduction of the vapor pressure difference between the fruit and the internal atmosphere. It affects the synthesis of volatile compounds (such as acetaldehyde, ethanol, and ethyl acetate) and prevents the development of off-flavors during storage [8].

### 4. Conclusions

The results showed the all US and packaging treatments significantly improved the quality of arils compared to control. However, a combination of US and VP is more effective than US and passive MAP in maintaining the overall quality of pomegranate arils during storage. The combination of US and VP can be a suitable alternative to other processes due to its non-destructive effects on pomegranate arils. However, more studies are necessary for process optimization.

**Author Contributions:** F.M.: methodology, writing, reviewing and editing. A.H.: methodology, formal analysis, writing. E.A.: methodology, investigation, formal analysis, writing. All authors have read and agreed to the published version of the manuscript.

**Funding:** Authors would like to thank gratefully the University of Birjand for providing the financial support of this project. Grant number 1617165.

**Institutional Review Board Statement:** The authors will follow the Ethical Responsibilities of Authors and COPE rules.

**Informed Consent Statement:** Not applicable.

**Data Availability Statement:** All data are presented in the manuscript.

**Conflicts of Interest:** The authors declare that they have no conflicts of interest.

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
