# Peer review of "Effects of Ultrasonication and Modified Atmosphere Packaging on the Physicochemical Characteristics and Quality of Ready-to-Eat Pomegranate Arils"

_horticulturae, doi:10.3390/horticulturae9070809_

Round 1

Reviewer 1 Report

I have reviewed the manuscript entitled "Synergistic effects of ultrasonication and modified atmosphere packaging on the physicochemical characteristics and quality of ready-to-eat pomegranate arils." This article presents relevant information about combining two technologies to preserve the quality of pomegranate arils. The manuscript is interesting and with valuable results. However, it is necessary to improve grammar and redaction throughout the manuscript, present the results considering the statistical analysis (factorial), and discuss the results based on the US, MAP, and VP treatments. Therefore some significant changes need to be made before being accepted for publication.

Specific comments:

1. Remove the term "synergistic" from the title, as the manuscript does not specifically address or discuss the synergistic effects of the combined technologies.

2. On line 17, indicate in the abstract what types of packaging.

3. On line 26, indicate what the abbreviation MAP means.

4. The abstract stated that the optimal treatment for certain variables was determined to be 15 min US + VP. However, in terms of flavor, VP alone yielded better results. This discrepancy may potentially confuse the reader. Therefore, I suggest emphasizing the treatment that performed better overall and providing a detailed explanation for this choice.

5. In the introduction section, lines 51-53, a verb is missing in the sentence "Various postharvest techniques, such as modified atmosphere packaging (MAP) and non-destructive ultrasound (US) method (ADD ARE) considered by many researchers".

6. In the introduction, it is essential to mention previous attempts made to preserve pomegranate arils that were unsuccessful. This will provide a basis for exploring alternative treatment options.

7. Enhance the writing style of the entire introduction. Describe each technology - ultrasonication (US), vacuum packaging (VP), and modified atmosphere packaging (MAP) - along with their respective advantages and disadvantages.

8. Existing studies have demonstrated the efficacy of these technologies in preserving whole and minimally processed fruits, including pomegranate. Therefore, it is crucial to highlight the novelty of your research. Explain why combining these technologies is of particular interest, despite their individual effectiveness.

9. In line 89, the objective is described, but it only mentions the US and MAP technologies. Please, include VP.

10. In the methodology section, on line 100, there is a mention of passive MAP. Clarify the meaning of the term "passive" concerning this technology.

11. On line 105, remove the conjunction "and" and rephrase the sentence to form a separate statement.

12. In section 2.2 of the methodology, it is unclear how fruit decay was assessed. Was it determined by observing the presence of fungi and the darkening of each aril? The term "infected fruits" is mentioned in the formula. Please specify whether this refers to the whole fruit or an individual aril. It should be noted that the study assumes the usage of ready to eat fruits.

13. In section 2.3 of the methodology, improve writing to ensure a clear understanding that positive values of "a*" represent red coloration, while negative values of "a*" correspond to green coloration. The same clarification should be made for the "b" values.

14. I recommend including another antioxidant technique in the methodology, such as FRAP or ABTS.

15. In section 2.9, include details such as the number of panelists involved in the sensory analysis, whether they were trained or not, any specific test conditions, and other relevant considerations that were taken into account during the sensory evaluation.

16. Improve the wording of lines 178-181.

17. In general, the description of results should be improved. I considered that the most relevant aspect of the manuscript should be comparing the US, MAP, and VP technologies. Based on this comparison, the results should be thoroughly explained, which is currently lacking in the manuscript. Therefore, please describe the results comparing the effect of the technologies. Additionally, the emphasis on time-related data appears excessive, as it is evident that fruits deteriorate more over time due to their inherent nature. Instead, the results should emphasize the contribution of the combined technologies. Comparisons between the effects of US+VP at different time intervals and US+MAP at different time intervals should be made, accompanied by a discussion on how these technologies, based on their fundamental principles, contribute to improving the studied variables. Time could be blocked in the statistical analysis to facilitate the analysis and interpretation of the results.

18. Include "was" before "obtained" on line 186.

19. In all the tables, it is currently unclear which specific day the data corresponds to in the treatments and which treatments are associated with the data presented in the storage time section. To address this, I recommend creating more comprehensive tables that include the following information: time, treatments, and variables. This will provide a more precise overview of the data and facilitate the understanding of the experimental results. For example:

                                       Weight loss %                           Decay rate %

                                   0    7     14 days                           0    7     14 days

5 min US+ VP

5 min US + MAP

Or

Storage time          treatment           Variable 1            Variable 2

0                        5 min US+VP

7                        5 min US + VP

14                     5 min US+VP

20. In section 3.3, specify the expected values of L, a, and b for pomegranate arils and then explain how the treatments positively or negatively impact the color characteristics of the fruit based on those expectations.

21. In line 245, a comparison is made with a previous study stating, "the power, temperature, and time of the US had a significant effect on the color characteristics of strawberry extract." Provide a detailed explanation of if this effect was improved, maintained, or worsened.

22. Enhance the wording of lines 248-251 to improve clarity and comprehension.

23. In section 3.4, please explain the desired outcome of TSS (Total Soluble Solids). Clarify whether it should be maintained, decreased, or increased. The current section is confusing as it suggests an increase in TSS, which may indicate an increase in metabolism. Elaborate on how the treatments influence this variable and its impact on the overall quality of the arils.

24. On line 307, change decreasing to a decrease in…

25. Explain how US at low intensity and short time, improves the extraction of anthocyanins (line 323).

26. On line 340, explain how US reduces nutrient loss and improves antioxidant capacity.

27. To enhance the description of the results, it is suggested to explain each variable studied by comparing the US+VP treatment against the US+MP treatment, rather than solely focusing on the effect of time.

28. Are all the references included necessary? Please select the most representative ones.

Author Response

Dear respected reviewer,

Many thanks for your valuable comments and suggestions that improved the quality of our manuscript. We corrected the manuscript according to all reviewers' comments. Response to your comments file also attached.

Regards

Reviewer 2 Report

Synergistic effects of ultrasonication and modified atmosphere packaging on the physicochemical characteristics and quality of ready-to-eat pomegranate arils 

1.  The manuscript is not formatted in the horticulturae;

2.  In the keywords delete the word "Ultrasonication", because it is already in the title of the manuscript;

3.  In the Introduction the paragraphs are very long, it is necessary to reduce them, or divide them into 2 or more, as in lines 39-63;72-90;

4.   "The pH of fruit juice measured with a digital pH meter". Enter the model and country of the manufacturer of the PH device;

5.  The conclusion needs to be rewritten as it looks like a discussion of results.

Author Response

(The authors gave the same response as above.)

Author Response

(The authors gave the same response as above.)

Round 2

Reviewer 1 Report

I have reviewed the manuscript entitled "Synergistic effects of ultrasonication and modified atmosphere packaging on the physicochemical characteristics and quality of ready-to-eat pomegranate arils." The authors have addressed some of the recommendations, and the manuscript is now ready for publication.